# Role of Oxytocin and Vasopressin in Neuropsychiatric Disorders: Therapeutic Potential of Agonists and Antagonists

**DOI:** 10.3390/ijms222112077

**Published:** 2021-11-08

**Authors:** Valeska Cid-Jofré, Macarena Moreno, Miguel Reyes-Parada, Georgina M. Renard

**Affiliations:** 1Centro de Investigación Biomédica y Aplicada (CIBAP), Escuela de Medicina, Facultad de Ciencias Médicas, Universidad de Santiago de Chile (USACH), Santiago 9170022, Chile; valeska.cid@usach.cl (V.C.-J.); macarena.moreno@ubo.cl (M.M.); 2Facultad de Ciencias Sociales, Escuela de Psicología, Universidad Bernardo OHiggins, Santiago 8370993, Chile; 3Facultad de Ciencias de la Salud, Universidad Autónoma de Chile, Providencia 7500912, Chile

**Keywords:** neuropeptides, oxytocin, vasopressin, pharmacology, antagonists, agonists, neuropsychiatry disorders

## Abstract

Oxytocin (OT) and vasopressin (AVP) are hypothalamic neuropeptides classically associated with their regulatory role in reproduction, water homeostasis, and social behaviors. Interestingly, this role has expanded in recent years and has positioned these neuropeptides as therapeutic targets for various neuropsychiatric diseases such as autism, addiction, schizophrenia, depression, and anxiety disorders. Due to the chemical-physical characteristics of these neuropeptides including short half-life, poor blood-brain barrier penetration, promiscuity for AVP and OT receptors (AVP-R, OT-R), novel ligands have been developed in recent decades. This review summarizes the role of OT and AVP in neuropsychiatric conditions, as well as the findings of different OT-R and AVP-R agonists and antagonists, used both at the preclinical and clinical level. Furthermore, we discuss their possible therapeutic potential for central nervous system (CNS) disorders.

## 1. Introduction

In recent years, two neuropeptides (NPs) namely oxytocin (OT) and arginine vasopressin (AVP), have gained attention due to their role in neuropsychiatric conditions, such as depression, anxiety, autism spectrum disorder (ASD), attentional hyperactivity deficit disorder (ADHD), substance abuse disorder (SUD), etc. The development of novel pharmacological treatments for the aforementioned pathologies, is a relevant task since most of the current therapies exhibit low success rates and a time gap between the beginning of the treatment and the first reported positive outcomes by patients [1,2].

Both NPs are nona-peptide molecules, differing only in two amino acids and are synthesized in specific areas of the hypothalamus [3,4], the paraventricular (PVN), and supraoptic (SON) nuclei. In general terms, two types of neurons have been identified as OT and AVP immunoreactive positive in the rat brain [5,6], the magnocellular neurons that project to the neurohypophysis and lead to the secretion of both NPs into the bloodstream as hormones, and the parvocellular neurons that project to the midbrain and other brain areas [7,8]. These cells differ in their morphology, size, amount of NPs production, and electrophysiological properties [9,10]. In addition, AVP immunoreactive cell bodies are observed in the bed nucleus of the stria terminalis (BNST) and medial amygdala (MeA) in several rodent species [11].

The OT and AVP receptors (OT-R and AVP-R, respectively) have similar structures and behave as promiscuous receptors, exhibiting a relative selectivity for both NPs. They are G-protein-coupled (metabotropic) receptors and act through G_q_, G_i_ or G_o_ proteins [12,13,14]. Moreover, the OT-R and AVP-R are expressed in different areas of the central nervous system (CNS) [15].

Several studies have shown the importance of OT and AVP on stress, anxiety, cognition, and social behavior in both preclinical and clinical settings [16,17,18,19,20]. In addition, the possible involvement of both NPs in the physiopathology of a variety of neuropsychiatric conditions, has been documented. For example, altered plasma levels of OT, as well as OT-R polymorphisms, have been associated with ADHD [21,22] or ASD [23]. Furthermore, low levels of AVP in cerebrospinal fluid (CSF) have been suggested as an early biomarker of ASD [24], whereas social recognition impairment due to early life stress is associated with a dulled response of AVP release in lateral septum (LS) in rats [25].

Based on these precedents, in recent years there has been a great interest in the exogenous use and/or modulation of these NPs as possible pharmacological treatments for various neuropsychiatric pathologies. However, due to the pharmacological and physicochemical characteristics of OT and AVP, the use of these molecules has encountered diverse limitations, mainly related to the inherent characteristics of most neuropeptides (e.g., short half-life, metabolic instability, and scarce blood-brain barrier penetration) [26,27,28]. This has fueled the search, synthesis, and use of synthetic agonist or antagonist substances exhibiting more stable physicochemical properties, allowing the crossing of the blood-brain barrier, and showing a higher selectivity by OT-R or AVP-R. This review summarizes the role of OT and AVP in neuropsychiatric conditions, as well as the findings of different OT-R and AVP-R ligands, used both at the preclinical and clinical level. Furthermore, we discuss their possible therapeutic potential for CNS disorders. The database used to select publications for inclusion in this article was mainly PubMed, and the key words used for the search were oxytocin/vasopressin receptor, oxytocin/vasopressin agonist/antagonist, clinical/preclinical trial of oxytocin/vasopressin agonist/antagonist, V_1A_R and V_1B_R.

## 2. Receptors: Brain Distribution, Expression Regulation, and Signaling

The actions exerted by OT and AVP at the CNS are closely linked to the expression and regulation of different receptors located at the cell surface level, triggering intracellular signaling and allowing for both NPs to modify the cell function [29,30]. Due to the diversity of receptors and functions, it is essential to understand the multiple signaling pathways associated with their activation and the different responses generated in different cells and brain regions.

Two types of OT-R have been identified in the CNS. They are metabotropic receptors, containing seven transmembrane domains, that act through protein-coupled G_q_ or G_i_ [31,32]. The OT-R is expressed in different areas of the CNS, for example, the OT-R coupled to protein G_q_ is expressed in ventral tegmental area (VTA), nucleus accumbens (NAc), anterior cingulate cortex (ACC), central and basolateral amygdala (CeA and BLA, respectively), medial preoptic area (mPOA), and anterior and ventromedial hypothalamus, etc. [33,34,35]. The OT-R type G_i_ is abundantly expressed in peripheral organs and tissues, but highly restricted in the CNS, where it is only found in the subventricular zone, rostral migratory stream, and olfactory bulb [36]. The OT-R type G_q_ activation induces an intracellular pathway mediated by phospholipase C (PLC), generating inositol triphosphate (IP_3_) and 1,2 diacylglycerol (DAG) as the second messengers. IP_3_ mobilizes calcium from intracellular stores, and DAG activates protein kinase C (PKC), contributing to the phosphorylation of proteins. In turn, the activation of these receptors results in depolarization induced by decreased inward rectifying K^+^ (IRK) currents [37,38,39]. The G_q_ signaling plays an essential role in maternal behavior, social behavior, initiation of social contact, and trust [40,41,42].

On the other hand, the OT-R G_i_ activation induces an inhibition of the adenylate cyclase (AC) activity, decreasing the concentration of cAMP, which activates phosphatidylinositol-4,5-biphosphate 3-kinase (PI3K) and increases IRK currents [36,38]. The principal effect linked to this isoform, which is primarily coupled to the modulation of different types of ions channels (Ca^2+^, Na^+^, K^+^), is to contribute to the regulation of membrane excitability, synthesis and release of neurotransmitters, and synaptic plasticity [43]. Recent data obtained from the human OT-R crystal structure have identified a conserved receptor-specific coordination site for Mg^2+^, a potent allosteric modulator for agonist binding. This is the first mechanism to describe the positive allosteric modulation of a G protein-coupled receptor (GPCR) mediated by divalent cations [44]. Interestingly, it has been shown that the presence of a divalent metal is essential for the binding of OT to OT-R [45]. In this context, Alshanski et al. in 2021 [46] determined the amino acids involved in the formation of the OT-Cu^2+^ complex in the solution using crystallography and nuclear magnetic resonance (NMR), evidencing both the OT amino acids and their binding order, which serve as ligands for copper. These findings account for how Cu^2+^ binds to OT in aqueous media. Therefore, this evidence should facilitate the development of peptidomimetic compounds based on this complex and its interaction with the receptor.

Vasopressin receptors V_1A_ and V_1B_ are widely distributed in liver, smooth muscle vascular cells, pancreas, and CNS. As previously mentioned, the V_1B_R distribution in the brain is more restricted than the V_1A_R. Both receptors are coupled to G_q/11_ and their principal signaling is via PLC [47]. Peripheral V_1A_R is involved in the regulation of blood pressure [48]. It is important to highlight that the activation of V_1B_R in the pituitary gland provokes adrenocorticotropic hormone (ACTH) release [49,50], while the blockade of this receptor has been shown to have anxiolytic and antidepressant effects on mice and rats [51,52].

On the other hand, V_2_R regulates water reabsorption in the kidney and is coupled to G_s_. Therefore, when activated by AVP, the result is an augmentation of cAMP levels thanks to the AC activity [53]. Mammal brain V_2_R expression and its relationship with social behavior is controversial [54,55]. Regarding the role of brain AVP-R in neuropsychiatric conditions and social behaviors, the V_1A_R gene (*avpr1a*) variability associated with polymorphic CpGs (polyCpGs) and methylation in wild prairie voles have been related to the differences in V_1A_R abundance in retrosplenial cortex [56] and adult sexual behavior [57]. In addition, early life stress in mice modify AVP gene expression, increasing AVP levels in hypothalamic neurons leading to impairment in corticosterone secretion and stress coping behavior [58]. In the case of V_1B_R, less research has been made regarding the impact of DNA changes upon behavior. Nevertheless, comparing mouse strains revealed differences in the 5′ microsatellite region in the 5′ *avpr1b* promoter, which have an impact on the promoter activity in cultured cells [59] suggesting a role in differential stress responses between strains [60]. Interestingly, V_1B_R KO mice may give a better understanding of the role of this receptor in stress response and social behaviors. Initial research showed that mice lacking V_1B_R were deficient in both cognitive and behavioral tests, such as social recognition and aggression assessment [61,62].

## 3. Neuropsychiatric Conditions: OT and AVP Role

### 3.1. Depression, Anxiety, and Stress

Although anxiety disorders and depression have a distinct underlying neurobiological mechanism, due to the high degree of comorbidity between these types of disorders, the evidence proposes common neurophysiological mediators underlying both conditions [63]. Another highly linked process associated with these disorders is stress, defined as a non-specific biological response that alters homeostatic processes in response to external requirements allowing for the individual to re-establish his homeostasis, and consequently facilitating his adaptation to the environment [64,65].

In this context, the oxytocinergic and vasopressinergic systems in the CNS are associated with or influence processes implicated in depressive and anxiety disorders as well as those underlying stress, making these NP systems potentially relevant to the development, maintenance, and treatment of these conditions, although usually in opposing directions. Specifically, OT has been shown to exert anxiolytic and antidepressant effects [66,67]. In addition, there is clinical and preclinical evidence indicating that OT can also improve the symptoms present in depression, such as sleep disorders [68] and anhedonia [69]. Furthermore, it has been shown that OT promotes neuronal regeneration processes, rescuing the suppression of neurogenesis processes induced by prolonged exposure to stress episodes and glucocorticoids in the hippocampus of rats [68,70], an effect that might be linked to the antidepressant effect conferred by this NP.

On the other hand, evidence shows that AVP has anxiogenic actions and increases depressive behaviors. Therefore, it has been shown that rats over-expressing AVP in the PVN exhibit high levels of anxiety and a depression-like phenotype, which is normalized after a long-term treatment with the antidepressant paroxetine [71]. In line with this evidence, clinical studies have revealed that patients diagnosed with depression show an overexpression of V_1A_R in the PVN [72].

Consequently, restoring the homeostatic balance of these NPs using modulators of AVP-Rs or OT-R activity, could be a practical way for treating mood disorders.

### 3.2. Substance Abuse

Addiction or substance use disorder (SUD) can be defined as a chronic disease characterized by drug seeking and compulsive use, and loss of control in limiting intake, despite its negative consequences. Moreover, it is associated with the appearance of negative emotional states (e.g., anxiety, irritability, dysphoria, and anhedonia) when the subject does not have access to the drug [73,74]. The chronic use or abuse of drugs induces behavioral changes associated with neuroplastic and neurochemical changes in the reward circuit, stress brain related areas, and executive function [75,76], which are mainly linked to the dopaminergic transmission of the mesocorticolimbic circuit, depending on the specific mechanisms of action of the drugs abuse [77,78]. These neuroplastic and neurochemical changes can be persistent. Therefore, it is considered a pathology with a high risk of relapse during the treatment.

There is considerable evidence showing that exposure to drugs of abuse can alter both the oxytocinergic and vasopressinergic systems in several areas of the CNS. In this context, preclinical evidence suggests that acute psychostimulants such as cocaine induce an increase in OT levels in the dorsal hippocampus of rats, a region linked to learning and contextual memory involved in drug dependence [79,80]. In addition, the acute cocaine exposure in rats has been shown to increase plasma levels of AVP, suggesting an increase in the neurohypophyseal release into the bloodstream [81]. Supporting these findings, a decrease of AVP content has been observed in the hypothalamus after 4 days of cocaine treatment, suggesting that the increase in AVP release at the hypothalamic level is followed by a depletion of this neuropeptide content in this region [82]. Recent studies from our laboratory showed that amphetamine (AMPH) induced conditioned place preference (CPP) and produced a decrease in AVP content in the LS nucleus, which is interconnected with areas linked to the regulation of mood and reward [83,84]. Moreover, the chronic administration of opiates (e.g., morphine) has been shown to differentially alter OT expression in various brain areas, including a decrease of OT in SON and NAc, and an increase in VTA [85]. In turn, an increase in AVP gene expression in the amygdala has been associated with early withdrawal from chronic opioid exposure [86]. Indeed, clinical evidence has shown that the use of alcohol impacts the NP systems both at the central and peripheral nervous system. For instance, chronic alcohol exposure produces dysregulation of plasmatic OT levels in men, whereas acute alcohol exposure induces a decrease in plasmatic OT levels in women [87,88]. Moreover, prolonged alcohol consumption induces a decrease in AVP immunoreactive neurons in the hypothalamus in rats and humans [89,90].

These findings have prompted studies in which these endogenous NPs have been evaluated as a way to reverse the neurophysiological and behavioral changes observed in addiction. Therefore, pre-clinical evidence suggests that the administration of OT influences the development of tolerance, sensitization, and withdrawal symptoms and can modulate diverse drug-seeking and drug-taking behaviors related to the excessive use of drugs [91]. Specifically, it has been observed that the administration of high doses of OT induces a decrease in locomotor hyperactivity and stereotyped behaviors associated with exposure to psychostimulants such as cocaine [92] and methamphetamine [93]. This effect has been linked to a potential modulation of dopaminergic activity in key regions such as the NAc, VTA, and PFC [94,95]. It was shown that the administration of OT (1 µg/µL) in the NAc prevents the cocaine-induced increase in dopamine (DA) release in this region [96]. Similarly, other studies with psychostimulants show that icv (2.5 µg/µL) or ip (1 mg/kg) administration of OT prevents the methamphetamine-induced increase in DA release in NAc [93,97]. Moreover, similar effects have been observed using an icv administration of OT (1 µg/µL), which prevents the increase of DA release in NAc, following acute or chronic ethanol exposure [98]. On the other hand, pre-clinical research showed that icv administration of AVP prolongs tolerance and hypnotic effect of alcohol [98,99,100] and increases cocaine self-administration. Indeed, AVP-deficient animals did not develop cocaine sensitization, suggesting that AVP could participate in the addiction process [101]. Contrary to this evidence, our group recently demonstrated that the microinjection of AVP in LS impairs the expression of AMPH-induced CPP and that this effect is mediated by the activation of the V_1A_R in the LS, which could produce the inhibition of GABAergic projections to the VTA, increasing the inhibitory tone in this nucleus, and therefore inducing a decrease in the dopaminergic activity at the NAc [84]. Moreover, animal studies suggest that the blockade of V_1B_R (but not V_1A_R) decreases the high level of intake in dependent animals and diminishes the emotional response to a stressor during withdrawal for cocaine [86].

Indeed, this evidence suggests that the NPs system remains a possible and attractive pharmacological target for treating addictive behavior and all of the neural adaptation associated with the use and abuse of drugs.

### 3.3. Social Behaviors

Social behaviors include different types of social interactions between peers and the use of high complexity communication and cues between individuals that are species specific [102]. Social behaviors include recognition and preference between individuals, memory of known peers, and more complex social interactions such as maternal, aggressive, sexual, play, etc. The implication of OT and AVP systems on social behavior are not recent in animal studies [103,104,105,106,107]. Analyzing social behaviors is crucial for understanding and proposing the correct treatment for conditions that display social deficits such as ASD, ADHD, schizophrenia, etc.

#### 3.3.1. Social Interactions

Social interactions can include any type of direct interactions between two or more conspecific (i.e., ano-genital or body sniffing, allogrooming, boxing, chasing, etc.). Additionally, in general terms, studies have reported a pro-social effect of the OT system. In this regard, using OT administration (0.1 mg/kg) on awake rats increased social interactions, while it selectively elevated DA extracellular levels in NAc [108]. It is worth mentioning that DA levels in NAc regulate motivation and reward processes related to social stimuli [109,110]. OT modulates social reward in NAc [111] and the main nuclei within the reward circuitry displays OT-R [94,108]. Therefore, the rewarding properties of social interactions and DA release could be mediated, at least in part, by the OT system [112].

As aforementioned, OT-R and AVP-R have a similar structure showing promiscuous binding of both NPs [113,114]. An investigation focused on social communication measured by flank marking in Syrian hamsters demonstrated this overlap. OT or AVP icv injections promote flank marking, while the V_1A_R agonist but not the OT-R agonist elicited the same outcome [115]. Interestingly, when the α-melanocyte stimulating hormone (α-MSH) was used to selectively stimulate OT release, flank marking increased, and this response was completely blocked by V_1A_R antagonism [115].

Striking results were observed in Mongolian gerbils after neonatal exposure to OT, AVP or OT-R and V_1A_R antagonists. Sociality was measured in two time points (neonatal and juvenile). In addition, neonatal AVP administration enhanced sociality in males but not females in both time periods. Interestingly, when male pups that received neonatal AVP were placed outside the nest, parents showed higher parental responsiveness towards those pups than littermates. However, they were not returned to the nest by mothers faster than the other pups [116]. The male pups administered with neonatal AVP, but not females, showed higher juvenile social behavior immediately after weaning, suggesting that neonatal neuropeptide treatments modified both parental responsiveness and play behavior in juvenile individuals in a sex-dependent manner [116]. It is important to mention that neonatal exposure to OT, OT-R antagonist (L-368,899; OTA) nor V_1A_R antagonist (SR49059) caused a significant effect over parental responsiveness or juvenile play behavior. These results could reflect a delay in the maturation between OT and AVP systems in young gerbils and sex effects in this maturation.

#### 3.3.2. Aggression

Applying isolation results in aggression in mice. In this line, both OT and AVP have been studied as anti-aggressive ligands. In brief, when mice were isolated for 6 weeks and treated with OT and AVP ip injections, the hyper aggressive behavior was reversed in a dose-dependent fashion and accompanied by an increase of social contact [117]. The posterior analysis revealed that only a high dose of the selective V_1A_R antagonist SR49059 was capable of blocking the effect of OT, but not after the OT-R selective antagonist L-368,899 [117]. These results are in line with the previously mentioned promiscuity between the NPs and their receptors and suggest that activation of the V_1A_R appears crucial for the anti-aggressive consequence of OT.

Importantly, sex hormones seem to have an effect on the aggressive behavior induced by isolation. When male and female Syrian hamsters were isolated and tested 4 weeks later, social isolation induced aggression to a same sex intruder in both sexes [118]. Then, when binding of V_1A_R and OT-R was analyzed using autoradiography, the effects of isolation were brain area sex specific. For V_1A_R binding, a significant sex effect in BNST, mPOA, and anterior hypothalamus (AH) was reported. A higher binding in BNST was described in females compared to males, specifically in grouped ones, whereas mPOA and AH levels were higher in isolated males. In the case of OT-R, sex effects were only observed in the lateral hypothalamus (LH) and PVN. In detail, OT-R binding in LH was higher in females, but isolation did not have an effect. Opposite sex effects were reported in PVN [118]. Interestingly, isolated hamsters also display higher levels of social behaviors compared to grouped (control) animals, although this latter group displayed the highest levels of non-social behaviors such as locomotion, exploration, grooming, nesting, feeding, and sleeping.

Moreover, sex differences have been documented regarding the AVP system and aggression. For instance, when Syrian hamsters of both sexes were acutely administered with AVP in AH (0.9 uM) 5 min before social/aggression behavioral testing, AVP reduced aggression compared to the vehicle in females, whereas in males the results were in the opposite direction [119]. In summary, AVP impacts aggressive behavior in a sex-dependent manner. These results highlight the differences in the neural substrates of aggressive behavior between sexes.

The V_1B_R has been associated with social recognition and social motivation, as well as territorial aggression in males [62] and maternal behavior in females [61]. Competitive aggression (aggression against other animals for resource access) is impaired in *Avpr1b*^−/−^ mice. When the animals were exposed to food restriction and then tested in a competitive aggression paradigm with other mice, knockout animals performed higher attack latencies and fewer attacks than the wild type individuals [61]. Similarly, defensive behavior is altered in mice lacking *Avpr1b* expression. After 2 weeks of isolation before each phase, the animals were tested as the intruder (phase 1) and then 6 weeks later tested as the resident (phase 2). Interestingly, the knockout mice showed normal defensive avoidance behavior. However, the defensive attack behavior was impaired when they were the intruder. The same animals that were tested as residents exhibited normal attack latency, but showed lower attacks per test [61].

Similar results were observed with the *Avpr1b* KO lentivirus approach in mice. A low percentage of *Avpr1b*^−/−^ males execute attacks on an intruder compared to the wild type. In addition, the behavior is restored when the lentivirus carrying *Avpr1b* is administered in hippocampal area CA2 [120].

#### 3.3.3. Maternal Behavior

Maternal behavior includes several social behaviors towards the offspring, for example, maternal care (nursing and grooming pups), maternal motivation (pup retrieval), maternal aggression (defense against an intruder), etc. [121]. It is important to mention that both OT and AVP have been related to maternal behavior. For instance, genetic variation in a single nucleotide polymorphism (SNP) in the promoter of AVP gene in Wistar rats provokes two phenotypes, the high (HAB) and the low anxiety-related behavior (LAB). This variation causes elevated AVP synthesis and release in HAB rats, a feature that has been associated with aggressive behavior against an intruder. Specifically, virgin and lactating HAB females are more aggressive than the LAB rats, but at the same time express high levels of maternal behavior [122,123]. The same trend was reported in mice [124].

In female mice, the lack of expression of V_1B_R (*Avpr1b*^−/−^) resulted in reduced overall maternal aggression (attack latency and number of attacks) compared to the wild type females when pups are exposed to an intruder (adult male mice) [61].

The AVP system modulates maternal care, motivation, and aggression through divergent brain areas. For instance, AVP release is increased in mPOA, BNST, PVN, and CeA and activates V_1A_R in mPOA, BNST, PVN, and/or CeA during the display of maternal care and maternal motivation [121]. On the other hand, V_1B_R impairs maternal motivation, especially when activated in MPOA, but has opposite outcomes when activated in BNST [121].

On the other hand, the OT system has been associated with pro-maternal outcomes. For example, in female mice, a network (piriform cortex, left auditory cortex, and CA2 of hippocampus) associated with maternal behavior is enriched in OT-R [125] in comparison with female virgins and male mice. Similarly, an interesting and exhaustive study in 2015 analyzed brain areas, maternal behavior (retrieval of pups), and the effect of OT in mice (mothers versus virgin females), demonstrating that virgin females retrieved pups after both OT systemic injection and optogenetic stimulation of OT neurons in PVN, only 2 h after the treatment [126]. It is important to mention that virgin females that received saline started the retrieval at least 2 days after cohabitation with pups, suggesting that the mere presence of the young animals could activate the OT system.

As an illustration of similar activities in humans, a correlation between the OT-R gene variation and parental sensitive responsiveness was observed (with factors such as maternal education, depression, and marital discord controlled) in mothers with OT-R AA or AG genotypes, which were less sensitive to their toddlers than mothers with the GG genotype [127].

#### 3.3.4. Social Play Behavior

Social play behavior has a crucial role in the normal development of cognitive and social interactions in young animals, which has been demonstrated after social isolation [128,129]. Moreover, early social dysfunction is observed in ASD and ADHD. Research on the brain systems behind those early deficits are critical for the possible rescue of those deficiencies.

In particular, the pharmacological blockade of V_1A_R in LS has a dichotomic effect. Therefore, it increases social play in male, but decreases it in female rats [130]. Similarly, AVP mRNA expression in BNST is negatively correlated with social play in males [131]. Using OT-R and V_1A_R antagonists microinjection in LS in juvenile rats, the role of both NPs and the sex modulation of social play behavior was established. The blockade of V_1A_R increases social play behavior in males in both home cage and novel cage context [132]. In the case of females, the response was in the opposite direction, i.e., females microinjected with V_1A_R antagonists play less than the vehicle-treated females, in both home and novel cages [132]. When AVP was administered, both sexes played less than the vehicle treated rats.

The blockade of OT-R had no effects on social play behavior in males, and the same results were observed after OT microinjection. Nonetheless, it was reported that in females, OT increases social play time only in the novel cage [132]. These data support the notion that both NPs modulate social play in a sex and context specific manner.

#### 3.3.5. Social Behavior Dysfunction and Neuropsychiatric Conditions

It is known that individuals with ASD exhibit social dysfunctions related to social recognition and social reward [133]. In this context, it is interesting that, as shown in an observational study, OT serum levels are correlated with autistic symptomatology and compared to ADHD males, ASD patients revealed higher levels of OT, although these were not different from those of a control group [23]. In an extensive review, Modi and Young in 2012 summarized several animal models and approaches to analyze the impact of differential neurobiological factors in ASD, including the OT system. In brief, both OT and OT-R have been investigated in mice, specifically using knockout animals. In both cases, the use of behavioral paradigms indicates abnormal behavior in tests such as ultrasonic vocalization induced by isolation and social recognition [134]. Regarding the treatment, intranasal OT (as well as non-peptide small agonists of the OT-R) showed positive outcomes in OT KO mice [134].

ASD presents higher incidence and severity in males [135], and accordingly, social deficits studies have explored the possible mechanisms behind this sex difference. In male mice, positive correlations between OT-R and V_1A_R mRNA expression and the degree of social interaction have been reported in MeA and PVN, where high social male mice express higher levels of both receptors of mRNA than the low social mice [136]. Importantly, high correlations appear along with estrogen receptor α (ERα) mRNA in MeA and ERβ mRNA in PVN [136]. This study demonstrated the implications of both receptors in social interaction and the possible modulatory role of sex hormones.

Another neuropsychiatric condition observed especially in the young population is ADHD, and as in the case of ASD, social abilities are diminished, and sex differences are suggested [133,137]. In this case, OT serum levels have also been associated with this disorder. Therefore, children with ADHD either drug näive or with an additional conduct disorder exhibit a lower OT serum level as compared to medicated ADHD patients or healthy controls [21,138].

Moreover, variations in the OT-R gene have been associated with ADHD. When OT-R gen was sequenced, three SNPs were associated with social deficit in children with ADHD. Therefore, a strong correlation was described for the CT/TT genotype of rs4686302 and low facial emotion recognition task [22], suggesting that rs4686302 polymorphism could be a genetic indicator of social dysfunction in ADHD pediatric patients.

Borderline personality disorder (BPD) is a condition characterized by distinct dysfunctions on three key domains of socioemotional information (affect regulation, behavioral control, and interpersonal sensitivity). On the other hand, it is believed that disturbances of the OT system are involved, at least in part, in the dysfunctions observed in the social reward and empathy circuitries. In BPD patients, imaging studies have reported alterations in the activation of the reward circuitry in response to social stimuli compared to healthy volunteers [139]. Regarding OT and reward systems, several studies have reported OT effects in the processing of negative and positive salience related to faces [140]. Currently, a model of dysfunctional OT system has been proposed as part of the mechanisms underlying BPD. In brief, it is believed that genetic factors such as SNP in the OT-R gene promote low parental OT levels that provoke two main outcomes: Decreased reward response to the own child and low salience for baby’s sensory cues, both leading to low sensitivity to child with poor caring and cuddling. This deficient relationship between the parent and child promotes low child OT levels leading to poor bonding and insecurely attached children associated with genetic factors in relation with OT-R in the child [141]. Therefore, diverse research has focused on OT administration and social functioning improvement, since it is presumed that OT could reduce the attention bias to social threat cues, regulate poor affect regulation and poor social reward experiences, and recover maladaptive empathy [141].

All of the aforementioned evidence places the OT and AVP systems as prominent constituents in the physiopathology of a variety of CNS diseases. This has stimulated the search of pharmacological tools aimed particularly at OT-R and AVP-R, which appear as remarkably attractive targets, for the development of novel treatments for psychiatric disorders characterized by social dysfunction. Therefore, in the next section, we will focus on the analysis of the characteristics of agonists and antagonists of both NP systems and the possible polypharmacy between them.

## 4. Preclinical and Clinical Studies and Human Research with OT and AVP: The Good, the Bad, and the Ugly

The study of the alterations of the oxytocinergic system at the CNS level has shed light on the neurophysiological mechanisms underlying various neuropsychiatric conditions, many of them linked to alterations in social, anxiety, addictive, parental care, decision-making behaviors, etc. [80,142,143,144]. For example, it has been demonstrated that the use of high doses of OT reduces locomotor hyperactivity and stereotyped behaviors induced by cocaine and METH [92,97,145]. On the other hand, it has been reported that the use of small doses of OT in the CNS prevents the sedative and ataxic effects induced by ethanol, but does not alter the development of tolerance to this drug [146]. The neurophysiological mechanisms that explain the effects of OT upon the actions of drugs of abuse are linked to the fact that OT administration reduces the level of DA in different mesolimbic regions, including the NAc, a critical region for addiction processes [94,147].

Clinical studies in healthy humans have shown that the administration of OT induces a decrease in the subjective perception of a stressful situation, which is accompanied by a decrease in salivary cortisol levels [148,149]. Therefore, an increase in OT levels or an increase in the activity of the oxytocin system during stressful situations may serve to counteract stress-induced physiological effects and decrease anxious behaviors. In this context, new findings have shown that the use of OT, but not AVP, stimulates neuronal growth and rescues the suppression of glucocorticoid- or stress-induced neurogenesis in the adult rat hippocampus [70].

Regarding social behaviors, clinical evidence has shown that the administration of OT via the nasal spray increases recognition and trust in healthy subjects and psychiatric patients, suggesting that OT could be a potential therapeutic drug for disorders such as autism or schizophrenia, etc. [150,151]. In line with this evidence, preclinical studies have shown that the central administration of OT increases social behaviors in monkeys [152], but the neurobiological mechanisms underlying these behavioral changes are not yet fully understood. Interestingly, new evidence has shown that OT plays an essential role in the non-social cognitive process. Therefore, it was demonstrated that the central administration of OT reduces preferences for risky outcomes in the probability discounting task, an effect blocked using OT-R antagonists, expanding the classical function associated with the OT system [144].

Despite interesting findings and advances in the role of OT in social and non-social behaviors, the usefulness of synthetic OT, both as a research and therapeutic tool, is limited by its physicochemical properties [153]. For instance, after peripheral administration, OT has a short half-life ranging from 2 to 4 min in humans and exhibits a poor penetration of the blood-brain barrier [154]. In addition, OT has high affinity for the AVP receptor V_1A_R, which has been implicated in numerous OT-dependent social behaviors [118,155]. Therefore, in recent years, efforts have focused on creating new molecules that possess more favorable physicochemical characteristics, as well as longer half-life and penetration of the blood-brain barrier [156]. Moreover, there exists interest in creating novel molecules with higher affinity and specificity for the OT-R, which, in theory, should allow for more precise and safer treatments.

Contrary to the broad therapeutic potential of OT, medicinal applications of AVP are more restricted. Specifically, AVP is used mostly in clinical settings and/or life-threatening scenarios, for septic shock, blood pressure regulation, and cardiovascular homeostasis [157]. Considering the wide distribution of peripheral V_1A_R and V_1B_R (liver, blood vessel muscles, kidney, heart, platelets, adrenal cortex, pancreas) and the critical responses that can be triggered after the activation or blockade of these targets, there are no AVP therapeutic uses authorized yet for CNS disorders. Nevertheless, animal studies have been conducted with AVP, which has unveiled its potential as a therapeutic alternative for social behavior flaws. As an example, using mice as an autism animal model, it was shown that *Magel2* KO adult male mice exhibit social deficits that were correlated with inappropriate AVP activation in projection neurons to LS and optogenetic stimulation of AVP neurons or administration of AVP rescued social deficits observed in *Magel2*^+/−p^ [158]. Furthermore, after 1 week of medium dosage of intranasal AVP (IN-AVP; 0.5 IU/kg), the partner preference is disrupted in male prairie voles, while an increase of novel object recognition was observed in animals of both sexes, and females increased play behavior before sexual maturation [159]. In agreement with other studies, after sexual maturation the high dosage of IN-AVP (5.0 IU/kg) produced an augmentation of aggression in males [159].

Regarding research in humans, IN-AVP effects on social processing of male and female faces have been reported. In particular, after two sessions (from 2 to 7 days between them) with one dose of IN-AVP (20 IU or 40 IU) and a final test, young males observed males and females faces and were asked to rate them regarding “approachability” (social perception), “willingness to initiate a conversation with the person” (social motivation), and “attractiveness” (sexual potential) [160]. Day 1 ratings of “approachability” and “initiate conversation” were the highest in the 40 IU single men group, although both ratings were lower in the 20 IU group as compared to the placebo [160]. Amusingly, the difference between female and male ratings in attractiveness were higher in the coupled men. Importantly, both systolic and diastolic blood pressure were not impaired during the test, although the mean percent change in systolic pressure 60 min after the application of 40 IU was lower than observed in the placebo group.

Similarly, a randomized placebo-controlled pilot trial evaluated the IN-AVP administration in children with ASD and reported promising results. After 4 weeks of IN-AVP administration in 30 children between 9.6 and 12.9 years, the treatment resulted in improved social abilities as measured by the Social Responsiveness Scale [161]. In addition, an improvement in anxiety symptoms was observed. Importantly, AVP was well tolerated, and minimal side effects were reported. These results highlight the possible therapeutic potential of AVP for ASD.

As aforementioned, research regarding therapeutic uses of AVP for CNS disorders has recently started and is far behind from the OT therapeutic development. However, in the past few years, several pharmaceutical companies have been developing and testing AVP alternatives for CNS disorders, such as migraine [162] or autism [163] (see the next section).

## 5. Agonist and Antagonist of OT-R and AVP-R: Peptides and Non-Peptides Alternatives

### 5.1. OT-R

In general terms, OT-R agonists or antagonists with better physicochemical characteristics compared to OT have been developed, although the majority of these molecules still have a considerable affinity for AVP-R and can behave as agonists or antagonists of V_1A_R or V_1B_R.

Several peptide and non-peptide OT-R agonists have been described. Most of them have been used only at a pre-clinical level, since they can generate severe damage when administered peripherally. Carbetocin is an OT analog with agonist properties. Most of the therapeutic uses of this compound have focused on its peripheral benefits associated with the prevention of postpartum hemorrhage (see review, Chao and McCormack 2019 [164]). Interestingly, Zanos et al. in 2014 [165] reported that carbetocin administered peripherally inhibited the development of anxiety and depressive behaviors during morphine withdrawal in addition to improving social behaviors. These results are consistent with the inhibitory effect of OT on stress linked to addiction [93]. On the other hand, pre-clinical evidence shows that early postnatal supplementation with carbetocin has a beneficial effect on myelination, long-term intrinsic brain connectivity and behavior in a rat model of perinatal brain injury associated with inflammation, deficient myelination, and behavioral deficit [166]. In the BALB/cByJ mouse model of ASD-like social deficit, carbetocin administration did not elicit prosocial effects [167].

Lipo-oxytocin-1 (LOT-1) is a synthetic peptide derivative of OT, with agonist properties. Research suggests that the ip administration of LOT-1 in the CD157 knockout model mouse of the non-motor psychiatric symptoms of Parkinson’s disease, has a functional advantage in the recovery of social behavioral impairment. This analog rescued anxiety-like behaviors and social avoidance [168]. In line with this evidence, Cherepanov et al. in 2017 [168,169] showed that the administration of LOT-1 in CD38 knockout (CD38^−/−^) mice, linked to the lack of paternal nurturing in CD38^−/−^ sires, decreased social recognition ability and decreased sucrose consumption. A recovery of the behavioral parameters in the long term, in a range of no less than 24 h was observed. Therefore, these results suggest that OT lipidation may have useful therapeutic benefits, mainly associated with its long-term action in modifying social behaviors. Another agonist with interesting characteristics is PF-06655075 (PFI). This is a non-brain-penetrant peptide with increased selectivity for the OT-R. In 2016, Modi et al. [170] showed that both central and peripheral administration of PFI inhibited freezing in response to a conditioned stimulus using the conditioned fear paradigm. Interestingly, it was shown that peripheral PFI administration resulted in a sustained level in plasma concentration for more than 20 h, but did not induce a detectable level at the brain tissue level. As a result, the authors suggest that exposure to plasma or cerebrospinal fluid might be sufficient to evoke the described behavioral effects.

An additional OT analog is TGOT, which has been widely used in the in vitro experiments showing a clear selectivity for OT-R vs. V_1A_R [171,172,173]. In discrepancy with these in vitro findings, an in vivo study showed that TGOT administration in OT-R null mice induced recovery of social deficit in *Oxtr*^−/−^ mice consistent with a selective action of TGOT upon OT-R. However, TGOT also rescued the social deficit in *Oxtr*^−/−^ mice, suggesting that, despite its very low affinity for V_1A_ and V_1B_ receptors in vitro, TGOT was still active on these receptors in the in vivo experiments [174].

New research has focused on the potential therapeutic use of OT metabolites, mainly OT (4–9) and (5–9). In 2019, Moy et al. [167] used a behavioral test in the BALB/cByJ mouse model, which presents a social deficit similar to autism. Mice treated with the metabolite (5–9) showed no change in their social deficit, whereas the metabolite (4–9) induced an improvement in social preference in these animals in a dose-dependent manner.

Several OT analogues containing tetrazole moieties have been developed due to the higher metabolic stability conferred by the presence of tetrazolyl group(s). In particular, in 2007, Manturewicz et al. [175] analyzed the biological activities of 11 OT analogues using substitutions of the Gln^4^, Asn^5^, and Gly-NH_2_^9^ residues in OT with the acidic 5-tetrazolyl group or *N*-methylated tetrazole rings. The uterotonic activity was decreased in analogues of both tetrazole or *N*-methyl groups as compared to the OT activity, although the potency increased in the presence of Mg^2+^. Similar results regarding *N*-methylation in different positions were recently reported [176]. A potency reduction (1000-fold) and elimination of agonist activity were observed after *N*-methylation at Tyr^2^, Ile^3^, Asn^5^, and Gln^4^, respectively [176].

On the other hand, OT-12, a potent and long-lasting OT analog has been shown as a powerful OTR agonist with a promising anorexigenic activity. Incorporating fatty acid moieties onto the backbone peptide of OT, results in low in vitro activity with AVP-Rs, improvement in plasma half-life in mice compared to OT and carbetocin, and a significant reduction in body weight in a mouse model of obesity [177].

In addition to the peptide analogs, a family of compounds classified as non-peptide has emerged [178]. It should be noted that due to the chemical nature of these molecules and the little preclinical evidence that exists, they have not yet been tested as treatments at the clinical trial level. One of the non-peptide molecules that acts as a potent OT-R partial agonist is TC-OT-39, which can also act as a V_1A_R antagonist. Evidence shows that ip administration of TC-OT-39 in the BALB/cByJ mouse model induces an improvement in the social deficit exhibited by these animals, an effect very similar to that shown by carbetocin [167]. Another non-peptide specific OT-R agonist is LIT-001, which has high affinity and efficacy for human and mouse receptors. Furthermore, LIT-001 shows low affinity for V_1A_R or V_1B_R, where it behaves as an antagonist and an agonist, respectively. Peripheral administration of LIT-001 induces an improvement in social interaction deficits present in *Oprm1*^−/−^ mice, a mouse model of autism, as evaluated by the recovery of the number and duration of nose contacts, and the frequency of the following episodes [179,180].

WAY-267464 has been characterized as a non-peptide agonist specific for OT-R [28,181]. Moreover, in vivo pharmacological testing has shown that WAY-267464 exhibits an antagonistic activity against V_1A_R [182]. Intraperitoneal administration of WAY-267464 improved social deficits in male *Shank3b* KO mice, a model characterized by the OT system dysfunction associated with decreased social salience [183].

Most of the scientific advances regarding the role of OT in various behaviors and pathologies have been done thanks to the use of OT-R receptor antagonists, which also can be classified as peptide and non-peptide compounds.

Moreover, atosiban (ATO) is a potent peptidergic OT-R antagonist that possesses an affinity for V_1A_R. The most classic clinical use of this drug involves peripheral administration, which causes the inhibition of uterine contractions induced by endogenous OT and thus prevents premature labor [184]. In addition, Abdullahi et al. in 2018 [185] reported the effect of blocking OT-R by the administration of ATO on contextual fear memory consolidation and reconsolidation in male rats. In this case, it has been proposed that administering different doses of ATO post-training in the fear conditioning paradigm impaired contextual fear memory consolidation, in a dose-dependent mechanism. In turn, the same doses of ATO administered for fear memory reactivation did not impair the reconsolidation of contextual fear memories.

Furthermore, human studies have shown that IN administration of OT restores time perception during social interaction in socially less competent individuals, whereas the administration of ATO decreases time perception during social interaction in socially competent individuals. These results indicate that OT might be involved in mediating time perception in social interaction, which further supports the role of OT in social cognition [186]. Interestingly, it has been shown that intra-hippocampal administration of ATO and co-administration of a GABA receptor type A antagonist (GABA_A_R) prevent the anti-epileptic effect mediated by diazepam [186,187]. It is noteworthy that seizures are a comorbidity present in the most severe cases of autism [188]. This finding opens exciting research and therapeutic fields by associating the role of the OT and how the imbalance in this system could be linked to the epileptic seizures observed in extreme cases of autism [189].

Ornithine vasotocin or OTA, is a selective peptide OT-R antagonist, which has been used to study the role of the OT system in attachment behaviors between the mother and offspring before weaning. Intraperitoneal administration of OTA has been shown to induce a decrease in attachment behaviors of the pups to the mothers, associated with a decrease in serum OT concentration in the pups before weaning [189,190]. In another study linked to maternal behaviors, it was demonstrated that bilateral microinjection of OTA into the central nucleus of the amygdala (CNA) is sufficient to induce aggressive maternal behavior similar to those observed by chronic cocaine exposure during the gestational period [191,192]. In line with this evidence, the administration bilateral in NAc of OTA, induced a decrease in social preferences in females and male monogamous mandarin voles (*Microtus mandarinus*) in the social preference paradigm [193].

Recent studies using the probability discounting task have shown that OT decreases nonsocial risk-based decision-making. This effect was only observed with the icv administration of OT, and it was blocked by the icv administration of OTA. This evidence provides new insight into the role of the oxytocinergic system in complex cognitive processes, such as decision-making [144].

Regarding non-peptide OT-R antagonists, L368,899 has been used in studies aimed to understand the neurobiological mechanisms underlying aggressive behaviors. It has been shown that the ip administration of L368,899 cannot block the OT-mediated anti-aggressive effect, but this effect can be blocked by high concentrations of a selective antagonist V_1A_R (SR49059) [117]. Therefore, these results suggest that both OT-R and V_1A_R activation participate in the anti-aggressive effects of OT. Therefore, the search for molecules or drugs with dual or polypharmacological action opens a window to a more efficient pharmacological therapy in pathologies associated with aggressive behaviors. On the other hand, a study carried out in rhesus monkeys, to which L368,899 was administered peripherally, indicated that this molecule can penetrate the CNS and modify the maternal behavior of females, eliminating interest in the baby and sexual behavior [144,194]. Furthermore, OT-R antagonism by L368,899 induced a decrease in the amount of time the male rat spent with the female, while no differences in exploitative behaviors, nor in the amount of time spent with another rat were observed [195].

Nevertheless, not all of the studies with this antagonist have been linked to social behaviors. It has been shown that the hippocampus is susceptible to damage caused by prolonged exposure to stress, causing alterations in neuronal plasticity (see review, Eun Kim et al. 2015 [196]). In this context, it was demonstrated that IN administration of OT has a neuroprotective role in periods of stress, an effect that is entirely reversed by exposure to L368,899 [197].

The role of AVP in the amygdala, and its involvement in the generation of emotional-affective responses related to pain, has been studied in rats using both AVP-R and OT-R antagonists. Therefore, AVP administration in the amygdala induced an increase in audible and ultrasonic vocalizations and anxiety-like behaviors. These effects were blocked by a selective V_1A_R antagonist (SR49059), but not by the OT-R antagonist L-371,257. In fact, the administration of L371,257 had some facilitatory effects on vocalizations, and on this basis, it was postulated that the activity of OT receptors, allows the mediation of some inhibitory effects of the vasopressinergic system [198].

### 5.2. V_1A_R and V_1B_R

Different pharmaceutical companies have searched for AVP-R modulators that could serve as a treatment for neuropsychiatric disorders. Therefore, a number of drug candidates acting through the V_1A_ and V_1B_ receptors such as terlipressin and F180 (V_1A_R agonists); relcovaptan (SR 49059), SRX251 and YM218 (V_1A_R antagonists); d[Cha4,Lys8]VP and d[Leu4,Lys8]VP (V_1B_R agonists), nelivaptan (SSR149415) and ORG52186 (V_1B_R antagonists), etc. have been described. Unfortunately, many promising modulators have been already abandoned as a result of failures in preclinical or clinical studies [199,200].

For example, in 2002, Gal et al. [201] characterized the first selective, nonpeptide vasopressin V_1B_R antagonist, namely SSR149415 or nelivaptan ((2S,4R)-1-[5-Chloro-1-[(2,4-dimethoxyphenyl)sulfonyl]-3-(2-methoxy-phenyl)-2-oxo-2,3-dihydro-1H-indol-3-yl]-4-hydroxy-N,N-dimethyl-2-pyrrolidine carboxamide). Both in vitro and in vivo experiments showed higher affinity for animal and human V_1B_R rather than V_1A_R or V_2_R. Interestingly, acute administration of SSR149415 (3 mg/kg ip and po) showed strong anxiolytic effects in rats, as evaluated in the four-plate test. Moreover, the anxiolytic-like activity of the compound (10 mg/kg po) lasted for at least 4 h [201]. Furthermore, the anxiolytic-like effect remained for longer periods when SSR149415 was given chronically for 7 days (10 mg/kg po). However, doubled-blind placebo-controlled clinical trials (Clinicaltrials.gov identifiers: NCT00374166, NCT00361491, NCT00358631, and NCT01606384) showed that SSR149415 failed to demonstrate competence in the treatment of major depressive disorder (MDD) and generalized anxiety disorder (GAD) [202], leading to discontinuation of trials in 2008 [203]. Although SSR149415 was safe and well-tolerated, doses of 100 and 250 mg twice daily were not superior in potency to the placebo, showing only a minimal response on the cortisol activity. These results could suggest a low power of V_1B_R blockade in humans or possible compensatory mechanisms that may block the therapeutic effect [203].

Between 2012–2014, Abbot and AbbVie reported and patented eight oxindole derivatives compounds with high V_1B_R affinity and selectivity [204,205,206,207]. Importantly, only two of those molecules (nelivaptan or SSR149415 and (2S,4R)-1-[5-chloro-3-(2-methoxy-phenyl)-2-oxo-1-(thiophene-2-sulfonyl)-2,3-dihydro-1H-indol-3-yl]-4-hydroxy-pyrrolidine-2-carboxylic acid dimethylamide) have been reported to have potential antidepressant effects [208].

The V_1A_R antagonist, SRX251, was evaluated in aggressive behavior in male Syrian golden hamsters using the resident-intruder model of aggression. A dose-dependent reduction in aggression, measured by latency to bite and the number of bites without changing other social behaviors such as investigation or sexual motivation, was observed after oral administration of the compound. These effects persisted for over 6 h post treatment [209].

Promising clinical trials in recent years have reported the pro-social role of the AVP system in autism. For instance, balovaptan (RG7314 or RO5285119 or RAX5D5AGV6) a selective V_1A_R antagonist, was approved for ASD in 2019 by the European Medicines Agency (EMA, EMEA-001918-PIP01-15-M02) after several studies (Clinicaltrials.gov identifiers: NCT03504917, NCT02901431; NCT04049578; NCT01793441; 2019-0000989-38; 2017-004378-32) were performed in volunteers with ASD. As an illustration, in a phase II clinical trial involving young men (*n* = 223) with moderate to severe ASD and an intelligence quotient ≥ 70, the subjects were given oral balovaptan for 12 weeks [163]. Regarding the item Socialization (in the Vineland™-II Adaptive Behavior Scales), positive effects were reported with 10 and 4 mg. Importantly, those improvements were measured using written questionnaires. Therefore, it remains an open question if the effects of balovaptan have a real impact over social deficits as observed in ASD.

Another recent phase III, randomized, double-blind, placebo-controlled trial (NCT03504917) investigated the efficacy, safety, and pharmacokinetics of 10 mg of oral administration of balovaptan once a day in adults with ASD. Using the Vineland™-II Adaptive Behavior Scales 2-Domain Composite (2DC) Score, volunteers that received balovaptan showed higher scores after 24 weeks. However, the difference between baseline and after 24 weeks of treatment in the Vineland-II Communication Domain Standard Score was lower. In the case of the Standardized scores on the Adaptive behavior, after 12 and 24 weeks, the percentage of subjects that received balovaptan and scored 6 points or more compared to the baseline were 34.4% and 43%, respectively, although these percentages were not higher than those observed in the placebo group (42.1% and 48.4%).

An exploratory phase II study (Clinicaltrials.gov identifier: NCT02055638) examined the safety, tolerability, and activity of the V_1A_R antagonist (SRX246) against the placebo, in adults of both sexes with the DSM-5 intermittent explosive disorder (IED). Volunteers were asked to answer questionnaires at weekly schedules. The protocol used was an 8-week SRX246 (4 weeks at 120 mg bid and 4 weeks at 160 mg bid) or an 8-week placebo, followed by a 1-week washout. SRX246 was well tolerated with a low incidence of adverse effects. Nonetheless, to date, the complete results regarding the reduction in Total Aggression Score have not been reported.

Recently, a randomized experimental approach using a neutral-predictable-unpredictable threat test evaluated the potential anxiolytic effect of the V_1A_R antagonist SRX246 in humans. Healthy volunteers that went under unpleasant electric shock and consumed 300 mg of SRX246 in a randomized, double-blind, and counter-balanced manner reported a decreased anxiety-potentiated startle [210]. It is crucial to mention two conditions of this study. First, anxiety was defined and measured as “startle reflex” (eye-blink) and the 36 volunteers were healthy men and women without any reported anxiety or mood disorder. Therefore, the clinical relevance of SRX246 in patients remains to be elucidated.

Finally, another V_1A_R antagonist, SR49059 or relcovaptan has been the focus of animal research in relation to aggression. Therefore, the hyper aggressive behavior triggered by an isolation protocol was reversed by AVP ip injections, in a dose-dependent fashion and accompanied by an increase of social contact [117]. High doses of the selective V_1A_R antagonist SR49059 were able to block the pro-social effects of OT [117]. These results support the idea of ligands overlapping between NPs receptors and suggest that activation of the V_1A_R appears as crucial for both AVP and OT anti-aggressive outcomes [117].

In summary, several OT and AVP ligands have been developed with promising results in the treatment of anxiety, drug abuse, autism, and depression both in animal models and clinical setups. In particular, OTR agonists and/or antagonists of V_1A_R and V_1B_R seem to be a good option for the treatment of a variety of neuropsychiatric disorders or some associated symptoms. However, questions regarding efficacy and safety still have to be solved in order to definitely establish their therapeutic activity (Figure 1).

## 6. Concluding Remarks and Future Projections

Selectivity has been considered as an essential property when designing therapeutic drugs. This has been based on the idea that selective compounds will have minimal side effects and maximum efficacy. However, there is increasing evidence that suggests that drugs targeting multiple receptors may show better efficacy and safety profiles [211,212,213,214]. This has led to the development of the concept of polypharmacology, which refers to the ability of a molecule to simultaneously interact with multiple target proteins or receptors [215]. The basic biological concept underlying the search for multitarget compounds is that robust pathological phenotypes, such as those seen in psychiatric or neurodegenerative disorders, are often the result of a complex web of molecular events rather than changes in the function of a single receptor.

In this context, OT-R and AVP-R appear as exceptionally attractive targets for CNS polypharmacology. For example, (a) the endogenous ligands themselves as well as some synthetic drugs, exhibit a certain level of promiscuity upon the corresponding receptors; (b) these proteins participate (usually with opposite functions) in the regulation of multiple physiological processes and are involved in the physiopathology of a variety of CNS diseases; and (c) as other GPCRs, NP receptors likely possess multiple binding sites. Furthermore, the orthosteric site of OT-R and AVP-R is well characterized structurally. Therefore, the pharmacophoric requirements to bind to this site are known and could be used in the development of drugs with multiple targets. On the other hand, these pharmacological targets likely contain numerous allosteric sites, which, as demonstrated for other GPCRs, should exhibit a wide variety of shapes, sizes, physicochemical properties, and ligands [212,216,217,218]. These features offer an unusual opportunity to explore the chemical space in search of molecules that could fit into these cavities.

Perhaps due to the limited number of molecules that act on OT and/or AVP receptors, there are virtually no examples of promiscuous drugs with described effects on these GPCRs. Far from considering this scenario as a limitation, we see it as a great opportunity to enter a highly unexplored field of research with significant potential for the development of new CNS polypharmaceuticals [218,219,220]. Therefore, for instance, it is important to mention that balovaptan and nelivaptan have a greatly similar profile, antagonizing the three proteins, OT-R, V_1A_R, and V_1B_R, although with differential potency. In summary, these data support the notion that new multitarget drugs acting simultaneously on different neuropeptide receptors (and possibly upon other important targets such as monoaminergic receptors) could have relevant therapeutic potential for the treatment of complex CNS pathologies such as anxiety, depression or drug addiction. In this regard, several molecules with therapeutic potential have been developed and tested in both animal and human studies for CNS conditions, some of which act as promiscuous ligands for OT-R and AVP-R with distinct affinities and functional profiles (Table 1).

## Figures and Tables

**Figure 1 ijms-22-12077-f001:**
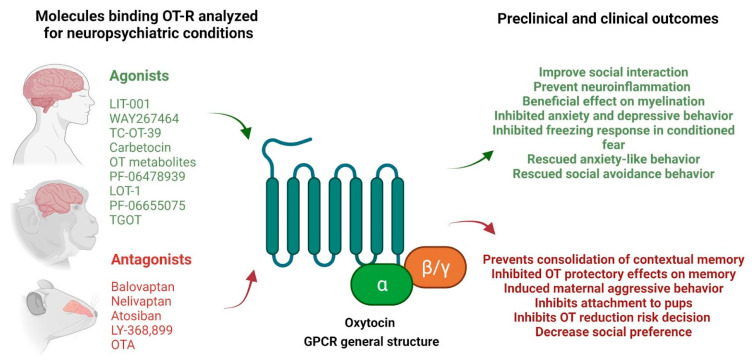
Examples of agonists and antagonists with affinity for OT-R and their principal preclinical and clinical outcomes reported. Scheme created with BioRender.com (October 19, 12:20 GMT-3).

**Table 1 ijms-22-12077-t001:** Summary of peptides and non-peptides molecules with polypharmacological profile in the OXT and AVP receptors in the central nervous system. Legend: (-) No-affinity or (np) not reported/(+) minimal affinity/(++) affinity similar to the endogenous ligand/(+++) affinity higher than endogenous ligand.

Molecule	OTR	V_1A_	V_1B_	Principal Outcomes Reported	Model/Disease/Condition	Refs.
RG7314 or RO5285119 or RAX5D5AGV6 or balovaptan	Antagonist (np)	Antagonist (np)	Antagonist (np)	Improvements in the Vineland-II socialization and communication scores.	Humans with ASD	[163]
SSR149415 or nelivaptan	antagonist (+)	antagonist (+)	antagonist (++)	Antagonized exogenous AVP-induced corticotropin secretion. Rats pretreated with nelivaptan induced inhibition in plasma corticotropin secretion and 30 min before the stress period caused a 50% inhibition of plasma corticotropin elevation.	Male Sprague Dawley Rats	[201]
				The forced swimming test produced a lower immobility time than the vehicle.	Wistar rats	[221]
				After a chronic mild stress test, degradation of the physical state of the animal’s coat was significantly improved by nelivaptan after 2 weeks of treatment.	BALB/c mice	[221,222]
				In generalized anxiety (GAD) and major depressive disorder (MDD), the treated patients did not show significant improvement from the baseline or did not separate from the placebo.	Patients with DSM criteria of GAD and MDD	[202]
SRX251	(-)	Antagonist (np)	(-)	Dose-dependent reduction in aggression	Male Syrian golden hamsters; resident-intruder model of aggression	[209]
SRX246	(-)	Antagonist (np)	(-)	Decreased anxiety-potentiated startle independent of fear-potentiated startle	Healthy volunteers	[210]
LIT-001	agonist (no-peptide) (++)	Antagonist (+)	Agonist to (-) high concentration	Improves social interaction. The activation of V_1B_R does not alter social behavior	Rat	[180]
WAY267464	agonist (+++)	Antagonist (+)	(-)	Improves social behavior in the SHANK-3B model Antagonism V_1B_Rfacilitates social interaction	Mice model of ASD (OPRM1 ^-/-)^	[183]
TC-0T-39	Agonist (+++)	(-)	(-)	Prosocial efficacy	BALB/cByJ model mice (ASD model)	[167]
Carbetocin	Agonist (+++)	(-)	(-)	No prosocial efficacy	BALB/cByJ model mice (ASD model)	[167]
				Prevent neuroinflammation-induced brain damage of perinatal origin Beneficial effect on myelination, intrinsic brain connectivity	Rat model of perinatal brain injury (low gestational protein diet LPD)	[166]
				Administered peripherally, inhibited the development of anxiety and depressive behaviors during morphine withdrawal	Addiction mouse model	[165]
Metabolites OT (4-9) and (5-9)	Oxytocin analogs	(-)	(-)	OT (4-9). Improves social preference, dose-dependent manner	BALB/cByJ (mice model of ASD)	[167]
PF-06478939	Agonist (+++)	(-)	(-)	Peripheral administration inhibited freezing in response to the conditioned fear stimulus	Rats, conditioned fear paradigm	[170]
LOT-1	Agonist (+++)	(-)	(-)	Rescued anxiety-like behavior and social avoidance in the open field test	CD157 knockout model mouse of the non-motor psychiatric symptoms of Parkinson’s disease	[168]
Atosiban	Antagonist (++)	Antagonist	(-)	OT-R. Prevents the consolidation of contextual memory to fear in rats	Rats, contextual fear memory paradigm	[185]
				V_1A_R. The microinjection into the hippocampus prevents the antiepileptic effect induced by diazepam		[187]
PF-06655075	Agonist (+++)	(-)	(-)	Decreased alcohol drinking	Rat model of alcohol dependence	[223]
SR49059 or relcovaptan	(-)	Antagonist	(-)	Inhibited vocalizations and anxiety-like behavior (elevated plus maze) in arthritic, but not normal, rats and conveyed anxiolytic properties to arginine vasopressin	Rats. Arthritic model (kaolin/carrageenan knee joint pain model)	[198]
				Inhibited anti-aggressive effects of OT in mice after 6 weeks of isolation	male Swiss mice	[117]
				Blocks the effect of OT in the inhibitory on METH-primed reinstatement of METH-seeking behavior	Male Rats trained to self-administer	[224]
LY371257	Antagonist (+++)	(-)	(-)	Facilitatory effects on vocalizations	Rats. Arthritic model (kaolin/carrageenan knee joint pain model)	[198]
LY-368,899	Antagonist (+++)	(-)	(-)	Reduced interest in the infant (primate maternal interest) and sexual behavior	Rhesus Monkeys	[194,195]
				Inhibited the oxytocin’s protective effects on hippocampal memory to stress	Model stress in rats	[197]
OTA	Antagonist (+++)	(-)	(-)	Induced maternal aggressive behavior	Female Rats	[191]
				Inhibits the attachment of pre-weaning pups to mothers	Male and female pre-weaning mandarin voles (*Microtus mandarinus*)	[190]
				Inhibits the role of oxytocin in selectively reducing risk decisions in male rats	Rats. Probability discounting task	[144]
				Decreased social preferences in volved in a dose-dependent manner	Female and male monogamous mandarin voles (*Microtus mandarinus*) using the social preference paradigm	[193]
TGOT	Agonist (+++)	(-)	(-)	Reduced isolation-induced aggression	Mice, social isolation induced aggression paradigm	[117]

## Data Availability

Not applicable.

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
