# Peer review of "Role of Oxytocin and Vasopressin in Neuropsychiatric Disorders: Therapeutic Potential of Agonists and Antagonists"

_ijms, 2021, doi:10.3390/ijms222112077_

Round 1

Reviewer 1 Report

I think this review of the literature is well written and have a potential role in the development of future studies. I think the authors have addressed clearly all the points included in the review and have summarized all the relevant areas of interest linked with these molecules. I have no concerns linked to this manuscript and I suggest the publication of it

Author Response

Thank you very much for your revision and comments.

Reviewer 2 Report

The review by Cid-Jofre et al. overviews the neuropsychiatric effect of Oxytocin and Vassopresin as well as agonists and antagonists for the receptor. The overview is well written and considers many aspects and works in the field. There are few things that can improve the review.

Here are my comments:

  1. The aspects of OT (DOI:10.1007/s00775-021-01897-1) and OTR (Ref 44 DOI: 10.1126/sciadv.abb5419), structures should be more highlight. This can help give an overview in agonist and antagonist design.  
  2. Agonists and antagonist molecular designs based on the peptide structure should be mentioned. For example, N-methylated (DOI: 1016/j.bmc.2016.05.062), Tetrazole (DOI: 10.18388/abp.2007_3169), and (DOI: 10.1021/acs.jmedchem.9b01862).
  3. The review lacks explanatory figures or schemes.

Author Response

Thank you very much for your suggestions and expertise.

1) The aspects of OT (DOI:10.1007/s00775-021-01897-1) and OTR (Ref 44 DOI: 10.1126/sciadv.abb5419), structures should be more highlight. This can help give an overview in agonist and antagonist design.  

R: Thank you very much for the suggestion. We included a paragraph in which the structure is highlighted, according to the reviewer`s suggestion and included the literature suggested (pages 4, lines 160-171).

2) Agonists and antagonist molecular designs based on the peptide structure should be mentioned. For example, N-methylated (DOI: 1016/j.bmc.2016.05.062), Tetrazole (DOI: 10.18388/abp.2007_3169), and (DOI: 10.1021/acs.jmedchem.9b01862).

R: According to the reviewer suggestion the oxytocin analogs using tetrazolic group or N-methyl substitution and OT-12 agonist, a potent and long-lasting oxytocin analog, were added in the text (Page 13, line 640-654).

3) The review lacks explanatory figures or schemes.

R: Thank you very much for this helpfully suggestion. We added an explanatory figure about agonist and antagonist oxytocin receptor and their functions (Figure 1).

Reviewer 3 Report

This review article provides an extensive overview of the role of vasopressin and oxytocin as well as of the agonists and antagonists of their receptors in neuropsychiatric disorders. I believe that it will be very helpful to researchers working in this field. Moreover, the concluding remarks and future projections section includes interesting suggestions for future research directions. I would like to propose several modifications:

  • Even though this is a narrative review, I suggest stating the criteria according to which publications were selected for inclusion in it. If this was based on a database search, please indicate the used databases, search words, and time period of the publications. If another approach was used, please indicate that.
  • I suggest presenting the information on the role of oxytocin and vasopressin on depression, anxiety, and stress in individual subsections. Even though, as the authors have stated, there is significant comorbidity, the underlying neurobiological mechanisms have also important distinctions.

Author Response

1)Even though this is a narrative review, I suggest stating the criteria according to which publications were selected for inclusion in it. If this was based on a database search, please indicate the used databases, search words, and time period of the publications. If another approach was used, please indicate that.

R: The database used to select publications for inclusion in this review was principally PubMed, the search words used were oxytocin/vasopressin receptor, oxytocin/vasopressin agonist/antagonist, clinical/preclinical trial of oxytocin/vasopressin agonist/antagonist, V1AR V1BR. These criteria were added in the text at the end of the section 1 (page 3, line 126-130)

2) I suggest presenting the information on the role of oxytocin and vasopressin on depression, anxiety, and stress in individual subsections. Even though, as the authors have stated, there is significant comorbidity, the underlying neurobiological mechanisms have also important distinctions.

We thank for the comment and absolutely agree with the reviewer. Nevertheless, our intention in this section is precisely to highlight that oxytocin and vasopressin might be part of the common neurochemical factors underlying anxiety and depression (and stress), and how this could guide the development of novel treatments for both type of conditions. We believe that this issue is explained in the second paragraph of this section. Despite these considerations, in the revised version we have included a sentence indicating that these disorders have distinct underlying neurobiological mechanisms (page 4, lines 199-201).